# Decorated Electrode Surfaces with Nanostructures and Metal-Organic Frameworks as Transducers for Sensing

**DOI:** 10.3390/s24206745

**Published:** 2024-10-20

**Authors:** Sara Caruncho-Pérez, Aida M. Díez, Ana Prado-Comesaña, Marta Pazos, María Ángeles Sanromán, Elisa González-Romero

**Affiliations:** 1Department of Analytical and Food Chemistry, University of Vigo, Campus As Lagoas-Marcosende, 36310 Vigo, Spain; ana.maria.prado.comesana@uvigo.es; 2CINTECX, University of Vigo, Campus As Lagoas-Marcosende, 36310 Vigo, Spain; adiez@uvigo.es (A.M.D.); mcurras@uvigo.es (M.P.); sanroman@uvigo.es (M.Á.S.)

**Keywords:** SPCE, nanoparticles, MOF, biochar, Clotiapine, Sulfamethoxazole, voltammetry

## Abstract

In this study, several materials are presented as modifiers of the screen-printed carbon electrodes with the aim of developing new sensing platforms for the voltammetric analysis of drugs. Specifically, Clotiapine and Sulfamethoxazole were selected as models for antipsychotics and antibiotics, respectively. Different nanostructures were studied as modifiers, including both transition metals and carbon-based materials. Moreover, biochar and two metal-organic frameworks (MOFs) were tested as well. The NH_2_-MIL-125(Ti) MOF showed an 80% improvement in the analytical signal of Sulfamethoxazole, but it partially overlapped with an additional signal associated with the loss of the MOF ligand. For this reason, several immobilization strategies were tested, but none of them met the requirements for the development of a sensor for this analyte. Conversely, carbon nanotubes and the NH_2_-MIL-101(Fe) MOF were successfully applied for the analysis of Clotiapine in the medicine Etumine^®^, with RSD below 2% and relative errors that did not exceed 9% in any case, which demonstrates the precision and accuracy achieved with the tested modifications. Despite these promising results, it was not possible to lower the limits of detection and quantification, so in this sense further investigation must be performed to increase the sensitivity of the developed sensors.

## 1. Introduction

Screen-printed carbon electrodes (SPCEs) are devices that integrate the three electrodes necessary for potentiostatic measurements on a ceramic or polymeric substrate. These devices maintain the high sensitivity, selectivity, and reproducibility characteristic of electrochemical techniques, but compared to the conventional three-electrode system, they are much more versatile. Due to their small size, it is possible to use them for decentralized analysis, as well as for the analysis of microvolumes without losing sensitivity, minimizing or even eliminating the generation of waste. Moreover, the ease of their large-scale production compared to other manufacturing methods makes them an affordable alternative to traditional electrodes such as glassy carbon and carbon paste [1].

The possibility of incorporating modifications onto the surface of the working electrode, makes SPCEs the ideal electrochemical transducers for sensing [2]. This is because the high sensitivity achieved in electroanalytical techniques is a consequence of phenomena that take place at the interface between the surface of the working electrode and the bulk solution. Therefore, the modification of this surface would entail changes in the interactions with the analytes under study. Specifically, the incorporation of certain modifiers can increase the catalytic effect by modifying the electron transfer rate or even the diffusion of the analytes towards the working electrode surface [3]. This catalytic effect can be reflected in an increase in sensitivity or even in selectivity by shifting the position of the analytical signal towards more favorable potentials, thus minimizing the possibility of interference.

Among materials frequently used as modifiers, nanostructures are prominent for their catalytic properties as a consequence of their high surface-to-volume ratio. Metal nanoparticles, specifically gold, have been widely used in recent decades for the analysis of both inorganic and organic compounds. These nanoparticles are also used together with other materials in composites that provide increased sensitivity and selectivity [4]. They are also commonly used to immobilize antibodies in nanoparticle-based immunosensors [5]. On the other hand, iron nanoparticles and their derivatives, such as iron oxide, are also used as modifiers because their magnetic properties enable them to immobilize substances through magnetic entrapment [6].

Although metal nanoparticles offer many advantages, in recent years carbonaceous nanostructures have been widely used for analyzing organic compounds because all organic molecules have a carbon backbone with functionalities that allow the establishment of dispersive forces and even electrostatic interactions [7,8]. Moreover, their large surface area and high electrical conductivity make them ideal for analyzing drugs from different families. Among the many carbonaceous nanomaterials, carbon nanotubes and graphene perform outstandingly in the field of quality control for both food and pharmaceutical formulations. In addition, they are frequently used in environmental analysis to analyze contaminants such as pesticides and drugs in water samples [9]. Another carbonaceous material that is regaining prominence is biochar, which is derived from biomass composed of condensed aromatic structures that have entrapped minerals [10]. In addition to presenting catalytic properties characteristic of carbonaceous materials, biochar is synthesized from pruning waste, which makes it cheap, eco-friendly, and a contributor to the circular economy. For this reason, biochar is increasingly used as a SPCE modifier as a cheaper and greener alternative to other carbon-based nanomaterials [11,12].

Apart from nanostructures, other materials have been reported as modifiers of SPCEs. Metal-Organic Frameworks (MOFs) are crystalline structures formed by metal clusters coordinated with organic ligands with semiconductor properties. These materials have high porosity and thus high surface area, in addition to the adaptability they offer by being able to combine different metals and ligands. In recent years, the excellent performance of these materials in fields such as environmental remediation and sensing, among others, has been demonstrated [13]. The most commonly used ligands are terephthalic acid, also named 4-benzenedicarboxylic acid (BDC), and its amino derivative 2-aminoterephthalic acid (ATA), and the most prominent metals are iron and titanium due to their redox activity and catalytic properties. By combining these materials, NH_2_-MIL-125(Ti) and NH_2_-MIL-101(Fe) MOFs have emerged. These MOFs are metal oxoclusters commonly used in water remediation, acting both as adsorbents and catalysts in the degradation of organic molecules, and their presence in the field of sensing as electrode modifiers, both alone and as composites, is also increasing [4,14,15].

The most commonly used reversible method for incorporating the modifiers onto the surface of the working electrode is drop-casting, since it is simple and relatively fast. However, sometimes the intermolecular forces involved are too weak to keep the modifier on the electrode, causing losses that can affect reproducibility and sensitivity [16]. Different immobilization strategies have been developed based on irreversible methods, especially physical entrapment with polymers. Several investigations have reported the use of polymers such as nafion and chitosan, but more complex methods like in situ electropolymerization have also been developed based on precursors including aniline, dopamine, and o-phenylenediamine (o-PD), among others [8,17,18,19]. Although these methods allow a large number of molecules to be trapped due to the three-dimensional structure of the polymers, their main limitation is the inhibition of mass transfer [20]. 

Another irreversible method is based on the formation of covalent bonds, notably the diazotization reaction. This is an irreversible method in which covalent bonds are formed due to the presence of aromatic amines that can react with nitrite in an acidic medium to form diazonium cations that are grafted afterwards onto the surface of the previously activated SPCE. The main limitation of this method is the requirement for exhaustive temperature control due to the low stability of the diazonium salts above 5 °C. In spite of this, this method is widely used due to the strength of the covalent bonds formed [21]. 

All the immobilization methods described are promising, but their effectiveness will ultimately depend on the analyte under study and its interactions with the modifiers used for its analysis. In these preliminary studies, the modification of commercial SPCEs is presented, and for this purpose, both affordable nanomaterials and MOFs were used in order to develop new sensing platforms. In addition, all the immobilization strategies cited above were compared. The developed sensing platforms were applied to the voltammetric analysis of two commonly prescribed drugs: the antibiotic Sulfamethoxazole (SMX) and the antipsychotic Clotiapine (CLT). Antibiotics, along with anti-inflammatories and anxiolytics, are among the most consumed drugs, which is evidenced by their growing presence in water around the world as emerging pollutants [22,23]. The consumption of antipsychotics is also increasing due to a greater occurrence of cases of depressive and sleep disorders, among others [24,25]. Therefore, not only is the quality control of their pharmaceutical formulations crucial, but their detection in water samples at trace and even ultra-trace levels is also necessary. However, while SMX has been widely studied in recent decades [26,27,28], the literature on the voltammetric analysis of CLT is sparse. Moreover, to our knowledge, no studies have been reported in which modified SPCEs are applied to the analysis of this drug, and so the research presented in this manuscript could open a new avenue of research for this compound. 

## 2. Materials and Methods

### 2.1. Reagents and Solutions

Clotiapine (8-chloro-6-(4-methylpiperazin-1-yl)benzo[b][1,4]benzothiazepine, MW 343.87 g mol^−1^) and Sulfamethoxazole (4-amino-N-(5-methylisoxazol-3-yl)-benzenesulfonamide, MW 253.28 g mol^−1^), both standard grade, were supplied by Sandoz (Basel, Switzerland) and Sigma-Aldrich (Burlington, MA, USA), respectively. Etumine^®^, the pharmaceutical formulation of Clotiapine, was supplied by Juvise^®^ Pharmaceuticals (Villeurbanne, France). Potassium ferricyanide K_3_[Fe(CN)_6_] (98%) and Na_2_SO_4_ (99%) were both obtained from Panreac (Barcelona, Spain). To adjust the pH and activate the SPCEs when convenient, different dilutions of H_2_SO_4_ (95–97%) from Fluka were used.

For the SPCE modification, carbon nanofibers (98% carbon basis) and magnetic iron oxide nanoparticles (0.5–0.7% iron basis), both supplied by Sigma-Aldrich, were used. Biochar, a zero carbon (particle size 0.5 mm) obtained by slow pyrolysis from Acacia-Australia, Acacia mimosa, and certified eucalyptus, was purchased from Ibero Massa Florestal S.A. (Ul, Oliveira de Azeméis, Portugal). For biochar activation, HNO_3_ (65%) supplied by Panreac was used. On the other hand, NH_2_-MIL-125(Ti) and NH_2_-MIL-101(Fe) MOFs were synthesized by the research group, and for this purpose, FeCl_3_∙6H_2_O (98%) and titanium (IV) isopropoxide (97%) from Sigma-Aldrich were used. 2-Aminoterephtalic acid (98%) and N,N-Dimethylformamide were purchased from Tokyo Chemical Industry and Supelco, respectively. Methanol (99.9%) and ethanol absolute were both supplied by Fisher Scientific (Waltham, MA, USA).

For the immobilization of the modifiers on the SPCE surface, the following polymers were used: Eastman AQ 55S from Eastman Chemical Products, cellulose acetate (acetyl content 39.8%), chitosan and nafion (5%) from Sigma-Aldrich. For the electropolymerization reaction, o-phenylenediamine from Sigma, and K_2_HPO_4_ (99%) and KH_2_PO_4_ (99%) from Panreac were used. The diazotization reaction was performed with NaNO_2_ (98%) and HCl 37% from Panreac. 

Solutions and suspensions were prepared in ultrapure water (Reverse Osmosis RO1-Compact/C system, 18 MΩ cm, Peter Taboada, Spain). All experiments were performed in Na_2_SO_4_ 10 mM as electrolyte at a pH value of 3. The 2.500 mM CLT standard solution was prepared in hydroalcoholic solution with an ethanol percentage of 50%. The SMX standard solution was prepared entirely in ultrapure water at a concentration of 0.605 mM. For this purpose, the solution was kept under magnetic stirring in the dark until complete dissolution, because SMX is specified to be photosensitive. Standard solutions were stored at 4 °C, and the vessel containing SMX was carefully covered with aluminum foil to protect it from light.

### 2.2. Instrumentation and Procedures

#### 2.2.1. Electrochemical Measurements

The voltammetric analysis was performed with a potentiostat-galvanostat PGSTAT30 (Autolab) equipped with a DropSens Connector for SPCE (Metrohm-Dropsens, Oviedo, Asturias, Spain) and controlled by software NOVA 2.1.2. The electrochemical cell consisted of an SPCE (Metrohm-Dropsens) composed of graphite working and counter electrodes and an Ag pseudo-reference electrode. The sample volume used to cover the electrodes was 50 µL in all experiments presented in this study. The commercial modified SPCE tested, supplied by Metrohm-Dropsens, were graphene (GPH), gold nanoparticles (AuNP), multi-walled carbon nanotubes (MWCNT) and carbon nanotubes-gold nanoparticles (CNT-AuNP). Furthermore, several materials were incorporated onto the unmodified SPCE, specifically magnetic iron oxide nanoparticles (FeNP), carbon nanofibers (CNF), biochar, and the MOFs NH_2_-MIL-125(Ti) and NH_2_-MIL-101(Fe). Since in all the experiments the electrodes used as a base were SPCE, the “SPCE/modifier” nomenclature will be used in the figure captions throughout the manuscript. 

The electrochemical processes that CLT and SMX undergo were studied by cyclic voltammetry (CV). The initial potential was 0.5 and −0.2 V for CLT and SMX, respectively. Both analytes were subjected to a potential sweep in the anodic direction up to 1.2 V, where the potential was reversed until returning to the initial potential. The step potential was 5 mV, and the scan rate used for the study of the number of scans was 100 mV s^−1^. For the scan rate study, this parameter was varied from 10 to 500 mV s^−1^. For the characterization of the modified SPCEs, the ferricyanide/ferrocyanide system, [Fe(CN)_6_]^3−/4−^, was used starting from the oxidized form [Fe(CN)_6_]^3−^ at a 5 mM concentration in 10 mM Na_2_SO_4_, and for this purpose, CV was applied in a potential range from 0.8 to −0.6 V with a step potential of −5 mV and a scan rate of 100 mV s^−1^. Electrochemical impedance spectroscopy (EIS) was also used for further characterization of some selected modifiers. EIS experiments were performed by applying a sinusoidal voltage perturbation of 10 mV in a frequency range from 10^5^ to 0.1 Hz using 10 points per frequency decade. In the same way as in CV characterization, 5 mM [Fe(CN)_6_]^3−^ in 10 mM Na_2_SO_4_ was used. 

The voltammetric technique selected for the analysis of both CLT and SMX was differential pulse voltammetry (DPV) because of its higher sensitivity compared to CV. The initial potential was 0.5 V for CLT and 0.2 V SMX, while the end potential was 1.2 V for both analytes. The step potential was 5 mV, the modulation amplitude was 50 mV, and the modulation time and interval time were 0.05 and 0.5 s, respectively, to get a scan rate of 9.9 mV s^−1^. These optimal conditions were previously studied to achieve the highest sensitivity with the unmodified SPCE. All experiments presented in this work were carried out at room temperature.

#### 2.2.2. MOF Synthesis

The synthesis of the NH_2_-MIL-101(Fe) MOF was based on the works of Li et al., as well as the research group led by Pazos and Sanromán [29,30]. Briefly, 1.7659 g of FeCl_3_∙6H_2_O and 0.5989 g of NH_2_-BCD were added to 40 mL of N,N-dimethylformamide (DMF), and the mixture was sonicated for 30 min in a Fisherbrand sonicator (FB 11203). Next, the solution obtained was transferred to a 40 mL autoclave, which was taken to an oven where it was heated at 110 °C for 20 h. The product was then centrifuged (for 10 min at 8000 rpm) in a sigma 3K30 centrifuge, and the supernatant was discarded. The solid was washed with water, ethanol, and DMF and after that, it was left to dry at 37 °C overnight. 

For the synthesis of the NH_2_-MIL-125(Ti) MOF, 1.0869 g of NH_2_-BCD was dissolved in 25 mL of DMF with magnetic stirring. Then 0.910 mL of titanium tetraisopropoxide was added to the solution, and 25 mL of methanol was added dropwise afterwards. The mixture was transferred to a 40 mL autoclave reactor to follow the same procedure as the NH_2_-MIL-101(Fe) MOF. After the product was centrifuged and the supernatant was discarded, the solid obtained was washed twice with DMF and methanol, and after that, it was left to dry at 37 °C overnight [31].

#### 2.2.3. Biochar Activation and Characterization

In order to increase the catalytic activity of biochar by incorporating acid groups, it was subjected to an activation process based on the literature [32]. Briefly, 2 g of biochar was mixed with 51 mL of HNO_3_ 65% and subjected to magnetic stirring at 700 rpm and 50 °C for 2 h. The resulting mixture was centrifuged at 3500 rpm for 10 min, and the solid was allowed to decant overnight. After carefully removing the supernatant, the solid was washed with ultrapure water. This centrifugation and washing process was repeated three times without the need to decant overnight. Finally, the resulting solid was dried in the oven at 120 °C for 3 h.

To determine whether the activation process was successful or not, both the raw and the activated biochar were characterized by elemental analysis (EA), Fourier transform infrared spectroscopy (FTIR), Raman, scanning electron microscopy (SEM), and energy dispersed spectroscopy (EDS). EA was performed by combustion chromatography with an elemental analyzer Carlo Erba 1108. FTIR was carried out with a spectrometer Nicolet 6700 (Thermo Fisher Scientific Inc., Waltham, MA, USA) coupled to a DTGS KBr detector and equipped with a KBr beamsplitter. The scans (32 scans per sample) were performed in a range from 400 to 4000 cm^−1^ with a resolution of 4 cm^−1^. Raman experiments were performed with a spectrometer Horiba Jobin Yvon HR800UV (Renishaw, Wotton-under-Edge, England, UK). Spectra were recorded with a laser of 633 nm in a range from 150 to 3000 cm^−1^, using a 50× objective with exposure times of 10 s. SEM and EDS were performed with a JEOL JSM 6010LA microscope (Jeol, Kanagawa, Japan) operating at high vacuum mode with an acceleration voltage of 10 kV.

#### 2.2.4. SPCE Modification

The modifications of the SPCE were performed by drop-casting 5 µL of the corresponding modifier previously dispersed in ultrapure water. These suspensions were prepared daily so as not to compromise the stability of the modifiers, especially the MOFs. Different drying strategies were tested, specifically drying at room conditions for 2 h, 6 h, and overnight. Oven drying and infrared light (infrared lamp Siccatherm 250 W, 230 V, Osram, Munich, Germany) were also tested for this purpose, but no significant differences were observed by these methods when compared to air drying. Conversely, the modifiers did not seem to fix more to the SPCE surface with longer waiting times, so in the end, 2 h air drying was maintained for all experiments. To promote the fixation of the materials to the SPCE surface, several polymers were tested, which were also added by drop-casting and dried at room conditions. Specifically, the polymers used were Eastman AQ 55S (AQ), cellulose acetate (CA), chitosan and nafion. The AQ solution was prepared by dissolving 300 mg in 1 mL of hot water to obtain a 30% solution that was diluted to 0.3% afterwards [33]. To prepare the ~1% CA solution, 102 mg of the solid were dissolved in a mixture composed of 6.4 mL of acetone and 5.3 mL of cyclohexanone [34]. This solution was kept under magnetic stirring for 1 h in a fume hood. The commercially available nafion was already a 5% solution, so it was simply diluted to 1%. Chitosan 0.5% solution was prepared by dissolving 50 mg in 10 mL of 1% acetic acid solution [17].

Another strategy tested to ensure the fixation of the modifiers was the in situ electropolymerization of o-PD. For this purpose, a solution containing o-PD 0.5 mM in PBS 50 mM at a pH value of 7.2 was prepared. The electropolymerization was performed by CV for 25 cycles, scanning at 50 mV s^−1^ from an initial potential of −0.4 V in the anodic direction up to 0.7 V, from which the scanning was reversed to −1.5 V [8]. This procedure was applied both before and after the drop-casting of the corresponding modifiers to check whether there was any significant difference between both strategies.

The diazotization reaction was also tested to immobilize the MOFs. Firstly, the SPCE was activated with 0.1 M H_2_SO_4_ by CV scanning from 0.5 to −1.1 V for 5 cycles at 100 mV s^−1^. Then, the SPCE was activated again with 25 mM HCl under the same conditions. After the activation step, the diazotization reaction was carried out by mixing 1 mL of MOF (taking into account the stoichiometry of the reaction, a 2 mM concentration of amine groups is needed) dissolved in 0.2 M HCl with 1 mL of 2 mM NaNO_2_ in an ice bath for 10 min, controlling the temperature so that it does not exceed 5 °C. Finally, the grafting was performed by CV using 50 µL of the resulting solution, applying a potential sweep from 0.5 to −0.7 V at 100 mV s^−1^ for 5 cycles [35,36]. This procedure was also carried out with ATA as a reference since it is the ligand of the MOFs.

#### 2.2.5. Modified SPCE Characterization

To characterize the modified SPCEs, the well-known [Fe(CN)_6_]^3−^/[Fe(CN)_6_]^4−^ system was used as a redox probe. It is a quasi-reversible system in which a single electron is exchanged, so studying the reversibility criteria is relatively simple. Overall, two parameters must be calculated which are the ratio between the anodic and cathodic peak intensity, i_p,a_/i_p,c_, and the separation between both peaks, E_p,a_—E_p,c_. When a reversible process is studied, the ratio between peak intensities must be equal to unity, while the separation between peaks must approach 59 mV per electron exchanged [17,37]. Therefore, values further from these criteria will be obtained when the electrochemical process becomes less reversible as a consequence of a slower electronic transfer.

#### 2.2.6. Real Sample Preparation

The medicine Etumine^®^ consists of pills with a CLT content of 40 mg per pill. An Etumine^®^ suspension was prepared by powdering one pill in an agate mortar until obtaining a very fine and homogeneous powder that was then dissolved in 25 mL of ethanol before diluting to a final volume of 50 mL with ultrapure water to obtain the suspension. Although the intention was to measure the medicine directly without any pretreatment, a positive matrix effect caused by a large amount of excipients implied a prior gravity filtration step, with which the matrix effect was minimized.

## 3. Results and Discussion

### 3.1. Characterization of Electrochemical Processes with Unmodified SPCEs

Before testing new sensing platforms, it is necessary to study the electrochemical processes involved with unmodified SPCEs by CV. Both SMX and CLT undergo a main oxidation process in which two protons and two electrons are exchanged, typical of organic molecules [28,38]. Specifically, SMX oxidation is quite complex, since a series of reactions take place through an electrochemical-chemical-electrochemical (ECE) mechanism. This is evidenced in Figure 1a, since two irreversible peaks can be observed that give rise to a quasi-reversible system when successive scans are applied. Conversely, in Figure 1b, it can be observed that CLT oxidation is a simpler irreversible process, since a single analytical signal appears in the voltammogram, whose peak intensity tends to decrease with the number of scans. Moreover, the analytical signal shifts towards more positive potentials, making the oxidation less favorable.

Furthermore, scan rate studies were also performed in order to determine the nature of the matter transport, and for this purpose, the Randles-Sevcik equation (Equation (1)) was applied, where i_p_ (A) is the peak intensity, n is the number of electrons transferred during the electrochemical process, A is the active electrode surface area (cm^2^), C is the analyte concentration (mol cm^−3^), D is the analyte diffusion coefficient (cm^2^ s^−1^), and *v* is the scan rate (V s^−1^).
i_p_ = 2.69 10^5^ n^3/2^ A C D^1/2^ *v*^1/2^(1)

When a linear relationship between i_p_ and *v*^1/2^ is obtained, the transport of matter is said to be governed by diffusion, while when the relationship is linear with *v*, there is an adsorption component that must be taken into account during the analysis. After applying Equation (1), a purely diffusive behavior was found for SMX [39]. In contrast, CLT (Appendix A) showed a significant adsorption component, following the equation i_p_ (A) = (1.9 ± 0.1)10^−6^ (A) + (3.04 ± 0.06)10^−5^ *v* (A s V^−1^) with a correlation coefficient R^2^ of 0.9990. These findings are in good agreement with the literature [40,41]. 

### 3.2. Modified SPCEs

After characterizing the electrochemical processes involved for the analytes under study, several modifications were tested in order to improve their analytical signal. For this purpose, DPV was selected based on the high sensitivity it offers for reversible and irreversible processes as well, as is the case of both CLT and SMX. Firstly, nanostructures were incorporated onto the SPCEs by direct immobilization. Among these nanoparticles, both metals and carbonaceous materials were used. Additionally, two NH_2_-MIL-type MOFs were also tested, and to ensure their fixation to the surface of the working electrode, different immobilization strategies with polymers were applied. Furthermore, diazotization was also studied as immobilization by covalent bond. Lastly, modifications with biochar and activated biochar were also tested. 

#### 3.2.1. Nanomaterials

In the present study, AuNP and FeNP were tested as examples of metal nanoparticles, since they are among the most widely used for the analysis of organic compounds. When it comes to carbonaceous nanostructures, the well-known GPH was studied, as well as MWCNT and CNF. The voltammograms obtained for these nanostructures are collected in Figure 2. Regarding the redox probe, the [Fe(CN)_6_]^3−/4−^ system shows a more reversible behavior with the carbon-based nanostructures, as seen in Figure 2a,b, meaning that the electronic transfer is favored by these materials. This is confirmed after applying the reversibility criteria, since in Appendix A it is shown that the ratio between peak intensities is closer to unity, while the separation between peaks decreases. Specifically, the most reversible response was obtained with the CNT-AuNP modification, probably due to the synergistic effect between these nanostructures. 

The voltammograms for SMX and CLT obtained with the modified SPCEs are reflected in Figure 2c,d, respectively. While for SMX, a catalytic effect does not seem to be observed with any of the modifications apart from GPH, an increase in the analytical signal of CLT is achieved with CNT-AuNP and MWCNT. GPH also showed an increase in the peak intensity of almost 60%, but unfortunately, the capacitive current also increased considerably. Bearing in mind that the signal-to-noise ratio must also be considered, GPH was finally discarded for both CLT and SMX. These data are collected in Appendix A. Therefore, CNT-AuNP was the material selected as a modifier for CLT, with an increase of the peak intensity of 54%, as well as MWCNT with a smaller increase of 32% for comparison purpose. No nanostructure showed an optimal response in the case of SMX, and therefore, other materials were tested to seek an enhancement on its analytical signal.

Before that, to study in more depth the more successful modifications, the characterization was also performed by EIS. Nyquist plots were obtained for each electrode material based on the equivalent circuit shown in Appendix A. This simple circuit is composed of an R_s_ component defined by the resistance of the measured solution, an R_ct_ component associated with the resistance to charge transfer, and a constant phase element CPE derived from the double layer capacitance. The presence of a Warburg element, which is a common component in diffusion processes, was also evaluated using Bode plot, but the results were not consistent with the presence of said element. 

The Nyquist plots for unmodified, and MWCNT and CNT-AuNP modified SPCEs are collected in Figure 3. For both modifications, a much smaller and less defined arc can be seen compared to the unmodified SPCE. While obtaining a semicircle indicates a significant resistive component, an almost linear behavior is related to purely capacitive responses [42,43]. Therefore, the modified SPCEs show less resistance to electron flow. In Appendix A, it can be seen that the R_ct_ value is lower for the modification with CNT-AuNP, which implies a greater electrical conductivity and thus an improvement in the catalytic performance, in agreement with Figure 2a. The R_ct_ value does not decrease for MWCNT, but the roughness factor α is lower compared to the unmodified SPCE. An α value close to 0.5 is associated with a non-uniform surface due to the presence of pores or flaws. Conversely, a value of 1 would correspond to a smooth or polished surface [42]. Specifically, a value of 0.98 was found for the unmodified SPCE, in agreement with the literature, while MWCNT and CNT-AuNP modified surfaces showed significantly lower values of 0.40 and 0.45, respectively [44]. Therefore, it can be concluded that both modifiers promote a rougher surface with more active sites available for the electrochemical reaction to take place, increasing the catalytic activity. This is consistent with the SEM images shown in Appendix A, where the greater porosity of the modified surfaces (Appendix A) can be seen compared to the unmodified SPCE (Appendix A), which shows a more uniform surface.

#### 3.2.2. MOFs

Two NH_2_-MIL-type MOFs, NH_2_-MIL-125(Ti) and NH_2_-MIL-101(Fe), were synthesized and used as modifiers of SPCE in a concentration of 2 mg/mL. For this purpose, these materials were dispersed in ultrapure water and sonicated until homogeneous suspensions were achieved before the modifications by drop-casting were performed. The voltammograms obtained with these materials are shown in Figure 4. In Figure 4a, it can be noted that the modifications were successfully incorporated, since the voltammograms obtained with the MOFs differ from the unmodified SPCE. The data extracted from the voltammograms for [Fe(CN)_6_]^3−^ are collected in Appendix A, where the increase in the reversibility after the modifiers were incorporated can be seen. 

In Figure 4b, a great increase of the peak intensity of SMX can be noted when the Ti-based MOF was used. Although the analytical signal was also enhanced with the Fe-based MOF, the improvement dropped from 81% with the Ti-MOF to 19%, as depicted in Table 1. Moreover, the half-width at half height, W_1/2_, increased significantly when using the Fe-MOF, which would imply a worsening in selectivity, because the wider a peak is, the easier it is to overlap with other signals in the presence of compounds that may be interferences. Conversely, CLT (Figure 4c) shows a significant improvement when using the Fe-MOF, since its peak intensity is enhanced by 42%, while the incorporation of the Ti-MOF inhibited the analytical signal. Thus, not only the CNT-AuNP and MWCNT modifications are worth investigating, but the Fe-MOF also seems to offer very promising results when it comes to CLT. 

For further characterization, EIS experiments were performed, and the corresponding Nyquist plots are depicted in Figure 5. The impedance is lower for the SPCE modified with the iron-based MOF, while with the Ti MOF, it increases compared to the unmodified SPCE. This could be due to the more pronounced loss of ligand in this MOF, which could act as a charge barrier for the negatively charged redox probe [45]. The R_ct_ value is also higher with the Ti MOF than with its iron analogue, which is in agreement with the lower catalytic effect shown in Figure 4a. Additionally, the roughness factor shows a less uniform surface for the modified SPCEs compared to the bare SPCE, which is consistent with the SEM images shown in Appendix A.

In order to evaluate the reproducibility of the modifications with the MOFs, five SPCE were modified with the Ti-MOF and Fe-MOF for SMX and CLT, respectively. For the former, a peak intensity of (12.3 ± 0.3) µA was found with an RSD of 2.6%, while for the latter, the peak intensity was (17.5 ± 0.2) µA with an RSD of only 1.2%. Given these results, it can be concluded that a reproducible coating can be achieved with these MOFs, at least as far as the analysis of the drugs studied in this work is concerned. Therefore, taking into account that three promising modifiers were found for CLT but none that met the all the requirements for SMX, the research focused on the latter from this point on. 

An aspect that must be considered with this modifier is the appearance of a shoulder that partially overlaps with the main peak of SMX. Although it does not seem to totally prevent the measurement of the analytical signal, it is notable that the new signal also appears in the blank. In the literature, it is stated that the Ti-MOF has low stability in water due to the release of the ligand, either by its exchange with water molecules or by hydrolysis leading to the breakage of the metal-ligand bond [13]. This could justify the appearance of the signal observed in the blank, which would imply that the MOF is not sufficiently fixed to the surface of the SPCE, since otherwise the release of the ligand would not be observed. 

To minimize the release of the ligand, it is recommended to previously activate the MOF by drying it in the oven between 150 and 250 °C. This is above the boiling point of water but below 300 °C to prevent the MOF from losing its crystalline structure [13]. Finally, a temperature of 150 °C was chosen to minimize alterations to the MOF. For treatment time, the activation was tested for 1 and 3 h for comparative purposes. The results show the redox probe (Figure 6a) does not seem to differ significantly, but the analytical signal of SMX (Figure 6b) worsens drastically after the Ti-MOF is subjected to the thermal treatment. Thus, in spite of the fact that the signal associated with the MOF does decrease slightly in the blank, the thermal treatment is not viable for using this modifier when analyzing SMX. Therefore, it is necessary to look for other ways to minimize the signal associated with the MOF without negatively affecting the peak intensity of SMX, and for this purpose, several immobilization methods were studied.

Firstly, several polymers were tested directly by incorporating them by drop-casting. The polymers studied for this purpose were the sulfopolyester AQ, the polysaccharides CA and chitosan, and the fluoropolymer nafion. In order to determine the effect of the polymers alone, the SPCEs were modified without the MOFs as references. Additionally, two different strategies were tested: first, adding the MOF and then immobilizing it with the polymers, and second, depositing the polymer before adding the MOF. While in the first strategy the MOF is immobilized by creating a layer of polymer on it, the second strategy is based on the electrostatic/dispersive forces that can be established between the polymer already fixed to the surface of the SPCE and the MOF. 

The voltammograms obtained during these studies are collected in Figure 7. The redox probe confirms the incorporation of the polymer in all cases. Specifically, with AQ (Figure 7a), CA (Figure 7c), and nafion (Figure 7e) the system becomes much less reversible, and the analytical signals decrease drastically until they become undetectable, as is the case with nafion. This confirms that the polymers have formed a layer that makes electronic transfer difficult. Specifically, AQ and nafion are two negatively charged polymers with which the also negatively charged redox probe suffers electrostatic repulsions, making it difficult to approach the modified electrode surface [33]. In contrast, chitosan (Figure 7g) increases significantly the peak intensities, since in acidic medium, it is a positively charged polymer that favors the electrostatic attraction of the probe [17]. Furthermore, the reversibility is also enhanced by this polymer. All this can be evaluated quantitatively, as shown in Appendix A. In the case of CA, determining the interactions established with the probe is more challenging because the polymer is neutral, and no electrostatic impediment is expected. Despite this, inhibition of the signals associated with the probe is observed. One possible hypothesis would be that the pore size of the CA film formed is smaller than the size of [Fe(CN)_6_]^3−^. This could explain the selective permeability of this polymer reported in the literature [34]. However, further studies are needed to confirm this. 

EIS experiments were also conducted to better determine the effect of the polymers on the SPCE, and the Nyquist plots are collected in Figure 8. The disappearance of the semicircular complex plane can be seen by the large increase in impedance caused by the negatively charged polymers AQ and nafion. Conversely, the SPCE modified with chitosan shows the lowest impedance due to the electrostatic attraction of the positively charged polymer with the redox probe. These findings are consistent with the CV experiments.

When the cited modifications were applied to the analysis of SMX, it was not possible to maintain the enhancement achieved with the MOF alone. The results were not satisfactory for AQ (Figure 7b) and CA (Figure 7d); in fact, with the latter, the analytical signal was inhibited in all cases. Conversely, there were improvements when using nafion (Figure 7f), even when the MOF was not added, but the peak intensity was lower than with the MOF alone. Furthermore, it can be seen in the voltammograms that the capacitive current is quite high, so this polymer is not the most suitable. This capacitive current could be due to the thickness of the polymeric layer, since this could make electronic transfer difficult. Lastly, chitosan (Figure 7h) did increase the peak intensity significantly when compared to the unmodified SPCE, but again, it could not compete with the MOF. These results are collected in Table 2. Notably, an improvement in selectivity can be inferred with AQ and nafion, since the peak of SMX shifts towards more negative potentials. This is due to the electrostatic attraction between the negatively charged polymers and the protonated SMX. In spite of this, and taking into account that a compromise between sensitivity and selectivity must always be sought, the polymers were discarded.

An in situ electropolymerization was then carried out using o-PD as a precursor, since it was the most viable among the candidates. Using dopamine and aniline was also considered, but in the end, both were discarded, the former due to the requirement of working in an inert and opaque atmosphere to prevent uncontrolled self-polymerization, and the latter due to its limited stability [18,46,47]. Electropolymerization was performed before and after the incorporation of the MOF, and for these studies, both the Ti and Fe-based MOFs were tested for comparison. Polymer growth can be confirmed by the appearance of peaks associated with po-PD, whose peak intensity increases with the number of scans, as shown in Appendix A in accordance with the literature [8].

The results obtained with the electropolymerization experiments are reflected in the voltammograms of Figure 9. The redox probe shows changes in peak morphology with both Ti (Figure 9a) and Fe-based MOFs (Figure 9c). The reversibility criteria were calculated for each case and collected in Appendix A. When analyzing SMX with the modified SPCE, broad and asymmetric peaks with high capacitive currents were obtained (Figure 9b,d), with no significant differences between both MOFs. Therefore, this strategy is also not optimal for immobilizing the MOFs, as the morphology of the analytical signal of SMX is negatively affected by the electropolymerization. Data extracted from these voltammograms are depicted in Table 3.

Lastly, diazotization was also tested, thanks to the presence of aromatic amines in the ligands of both MOFs under study. The diazotization was also performed with ATA, the ligand of the MOFs, in order to have a reference and thus enabling to compare the effect of the free ligand with the bonded ligand. The voltammograms obtained for [Fe(CN)_6_]^3−^ by the diazotization methods with the Ti and Fe-based MOFs are depicted in Figure 10a,c, respectively. Here, it can be appreciated that the peaks associated with the redox probe are completely inhibited after diazotization with both the ATA and the MOFs, as reflected in Appendix A. This inhibition, also reported in the literature, is due to the formation of an insulating monolayer that prevents the electronic transfer of the redox probe, confirming the successful grafting [48].

Conversely, when testing the SPCEs modified by this method with SMX, the analytical signal worsened with both the Ti (Figure 10b) and Fe-based MOF (Figure 10d). This decline was observed not only in comparison to the unimmobilized MOF, but also to the unmodified SPCE, as can be seen quantitatively in Table 3. Therefore, although this method successfully eliminated the signal associated with the MOF, meaning that the modifier was effectively fixed to the surface of the working electrode, it failed to enhance the analytical signal of the drug.

#### 3.2.3. Biochar

After testing several modifiers, a suitable sensor for SMX has not yet been achieved, as no material other than the Ti-based MOF significantly increased the analytical signal without causing a negative impact on other parameters, such as the capacitive current or the peak width. Unfortunately, although this MOF managed to improve the peak intensity between 75–90%, its low stability in water resulted in a loss of the ligand that generates a peak that overlaps with the SMX analytical signal. In spite of having tried different immobilization strategies to solve this problem, none of them achieved the enhancement in peak intensity observed with the MOF alone. For this reason, it was decided to test biochar as modifier, since it is a cheap carbonaceous material with an excellent electrical conductivity that could have affinity with the carbon skeleton of the drug [11]. 

Biochar was tested directly, and also after being subjected to a prior activation treatment to functionalize it with acid groups and thus favor its catalytic properties. In order to determine if the activation process took place properly, several characterization techniques were used. Firstly, it can be confirmed that the activation process has taken place successfully through EA, since in the activated biochar, there was 2.14% of nitrogen that was not present in the untreated biochar. This nitrogen is probably due to the remains of HNO_3_ that could not be eliminated during the washing. Moreover, EDS analysis showed an increase in oxygen from 4.08 to 17.48%. This significant increase in the percentage of oxygen corroborates the presence of acid groups derived from the activation. The confirmed presence of other elements, such as Si, K, and Ca, decreased after the activation process due to the acid treatment.

Otherwise, in the FTIR spectra of activated biochar, a band associated with the C=O bond stress appears at 1711 cm^−1^, bonds that would be expected in the acid groups incorporated in the biochar. This band did not appear in the spectrum of the untreated biochar, thus confirming the effectiveness of the activation treatment. Raman spectroscopy was also applied, since it is a technique widely used to study carbonaceous materials due to its ability to detect small structural changes usually associated with disorder [49]. When studying the relationship between the intensities of bands G and D, I_D_/I_G_, the greater this ratio, the greater the degree of disorder in the material under study [50]. In the case of biochar, the ratio between the intensities calculated using fitting with Lorentzian functions was 1.40 for untreated biochar and 1.25 for activated biochar. Therefore, untreated biochar presents slightly more disorder in its structure than after being subjected to the activation treatment. This disorder could be due to the presence of flaws or overall heterogeneity in the sample. FTIR and Raman spectra for both materials are reflected in Appendix A, respectively.

The conclusions drawn from the Raman spectra can be confirmed using the SEM images depicted in Figure 11. Here, it can be noted that, although both materials present clear flaws in their structures, the untreated biochar background appears more heterogeneous (Figure 11a), while the activated biochar shows a more integrated structure (Figure 11c). Additionally, in Figure 11d, it can be noted that activated biochar seems to have better defined pores than the untreated material, as seen in Figure 11b. Greater porosity implies an increase in the surface area, so a greater catalytic effect would be expected with activated biochar. 

Once the materials had been characterized to confirm the effectiveness of the activation process, both of them were used for the modification of the SPCE, focused on the analysis of SMX. Furthermore, due to the low water solubility of these materials, they were immobilized with the polymers previously studied with the MOFs to avoid losses. This time, only the strategy where the modifier is first drop-casted, and the polymer is added afterwards, was tested, as it yielded the best results with the MOFs. 

The results obtained for SMX using biochar as a modifier are collected in Figure 12, where it can be noted that, both for untreated (Figure 12a) and activated biochar (Figure 12b), the best response in terms of the peak intensity is achieved when the modifier is immobilized with nafion, which can be confirmed with the data collected in Table 4. In spite of this, this polymer is again the one that generates the greatest capacitive current. On the other hand, when comparing the results obtained with the activated biochar with the untreated material, no significant differences are observed beyond immobilization with chitosan. This demonstrates that the treatment to which the material was subjected does not seem to be enough to improve the analytical signal of SMX. 

### 3.3. Determination of Clotiapine in Etumine^®^ with Modified SPCEs

While the most suitable sensor for SMX was not determined with the tested materials, three candidates were found for CLT, specifically MWCNT, the MOF NH_2_-MIL-101(Fe), and CNT-AuNP, which offered a signal improvement of around 30, 40, and 50%, respectively. In view of these results, it was decided to carry out the analysis of CLT in a real sample, specifically the medicine Etumine^®^, with the modified SPCEs. For this purpose, studies of the effect of concentration were performed to determine the linear range, as well as the Limits of Detection (LOD) and Quantification (LOQ). 

The voltammograms obtained during these studies are reflected in Figure 13, where it can be seen that more narrow analytical signals are achieved with the unmodified SPCEs (Figure 13a) and after the incorporation of the MOF (Figure 13b), but when the SPCEs are modified with MWCNT (Figure 13c) and CNT-AuNP (Figure 13d), the peaks become slightly wider. Therefore, this effect is expected to be due to the carbon nanotubes themselves, since they are factor common to both modifications. The capacitive current also seems to increase with these modifications. In contrast, the peak potential of CLT when using unmodified and Ti-modified SPCEs was about 0.83 and 0.81 V, respectively, while with both the MWCNT and CNT-AuNP, the peak shifted towards a more favorable potential of 0.76 V. Bearing in mind that the modifications with which a more favorable peak potential is achieved are those in which the half-peak width increases, it is difficult to infer a compromise in terms of selectivity.

Sensitivity would be expected to increase with the three modifications, since the previous studies showed a significant increase in peak intensity of CLT compared to the unmodified SPCEs. To evaluate this, the corresponding analytical parameters listed in Table 5 were calculated. LOD and LOQ were determined from the equations (3 × S_a_)/slope and (10 × S_a_)/slope, respectively, where S_a_ is the error of the intercept. Comparing the slopes, the increase in sensitivity achieved with the modifications is clear. However, the correlation between the analytical signal and the concentration slightly worsens, and the error in the intercept is greater than when the SPCEs are not modified, resulting in slightly higher LODs and LOQs. This is probably because the instrumental conditions applied for these studies were previously optimized for the unmodified SPCE, but not for each specific modifier. As previously mentioned, the studies reflected in this manuscript are preliminary and explanatory, with a more illustrative and comparative objective. To determine whether these modifications are actually valid to improve the quantification of CLT or not, more comprehensive studies must be performed to optimize the method for each specific case. In spite of this, it can be concluded that limits of the order of micromolar can be reached when analyzing CLT with MWCNT, CNT-AuNP, and the MOF NH_2_-MIL-101(Fe), thus confirming the high sensitivity achieved with these modifications.

Once the linear dynamic range of CLT was established for each modification they were applied to the analysis of the pharmaceutical formulation Etumine^®^. For this purpose, samples were prepared in triplicate, and convenient dilutions were made to interpolate on the calibration lines obtained during the concentration effect studies. The CLT content found in the medicine by DPV with the tested modifications is depicted in Table 6, where it can be observed that in no case does the relative error exceed the 10% stipulated by pharmaceutical companies as acceptable, so it can be concluded that the results are accurate. Specifically, the best response in terms of accuracy is achieved with the unmodified SPCE, while the worst was obtained with the Fe-MOF. This is logical, since unmodified SPCEs are manufactured by specialist companies, while the modifications with the Fe-MOF, although proven to be reproducible, cannot be compared with a large scale thoroughly controlled manufacturing process. Conversely, the RSDs calculated were excellent in all cases, which demonstrate the high precision that can be achieved using DPV with the studied modifications. 

## 4. Conclusions

In this research, several materials were tested as modifiers of SPCEs to develop new sensing platforms aimed at enhancing the analysis of SMX and CLT. These materials included nanostructures, specifically metal nanoparticles and carbonaceous materials. Additionally, biochar, subjected to a chemical activation process to functionalize its surface and enhance its catalytic properties, was also tested. Moreover, two NH_2_-MIL-type MOFs were studied. 

It was confirmed that all the modifications were successfully incorporated onto the surface of the working electrode of the SPCEs by studying the reversibility criteria of the [Fe(CN)_6_]^3−/4−^ system used as a redox probe. However, only one material, the NH_2_-MIL-125(Ti) MOF, significantly increased the analytical signal of SMX, with an enhancement up to 80%. Unfortunately, this increase in peak intensity was accompanied by the appearance of a signal associated with ligand losses, that partially overlaps with the SMX signal. To overcome this problem, different immobilization strategies were tested, but although they made it possible to minimize and even eliminate the signal associated with the MOF, they could not match the performance achieved with the modifier alone. Therefore, more materials and modification strategies must be sought in order to find the ideal modifier for SMX.

In contrast, for the voltammetric analysis of CLT, three effective candidates were identified as modifiers: MWCNT, the NH_2_-MIL-101(Fe) MOF, and CNT-AuNP, which improved the analytical signal by 30, 40, and 50%, respectively. These three materials were applied to the analysis of CLT, and after determining the dynamic linear range and calculating the LOD and LOQ, which in all cases were of the order of the micromolar, the quality control of the medicine Etumine^®^ was performed. The results obtained were accurate and very precise with relative errors and RSD below 9 and 2%, respectively, thus confirming the viability of the studied materials as SPCE modifiers in the quality control of medicines. Although these results seem very promising, further investigation must be carried out to optimize the method adapted to each modifier and to achieve a significant increase in sensitivity that really has an impact on the analysis of this drug.

## Figures and Tables

**Figure 1 sensors-24-06745-f001:**
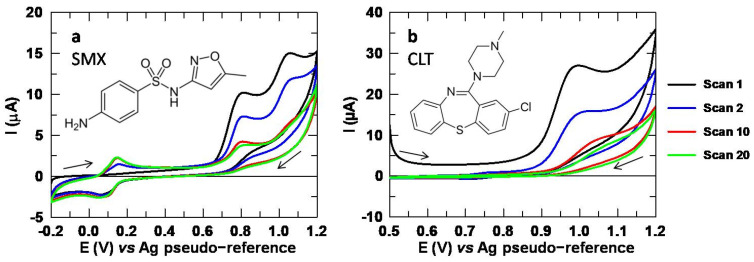
(**a**) Voltammograms obtained by CV in the anodic direction of SMX 150.0 µM and (**b**) CLT 151.2 µM. Electrolytic medium: Na_2_SO_4_ 10 mM at pH 3.

**Figure 2 sensors-24-06745-f002:**
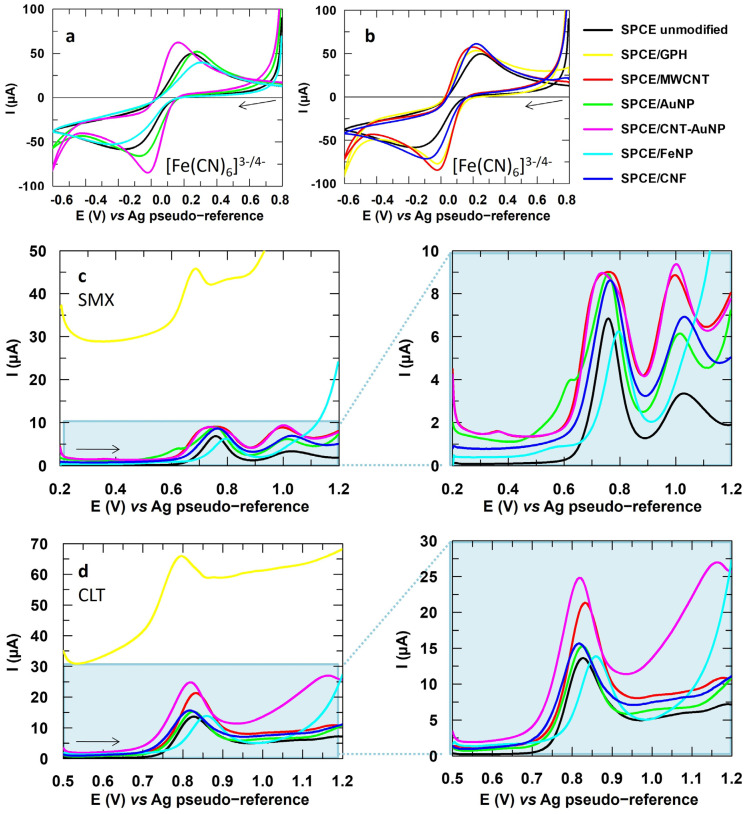
Cyclic voltammograms obtained for [Fe(CN)_6_]^3−^ 5 mM with (**a**) metal nanoparticles and (**b**) carbon-based nanostructures. (**c**) DP voltammograms obtained for SMX 151.2 µM and (**d**) CLT 150.0 µM. Electrolytic medium: Na_2_SO_4_ 10 mM at pH values of 6 for [Fe(CN)_6_]^3−^ and 3 for SMX and CLT.

**Figure 3 sensors-24-06745-f003:**
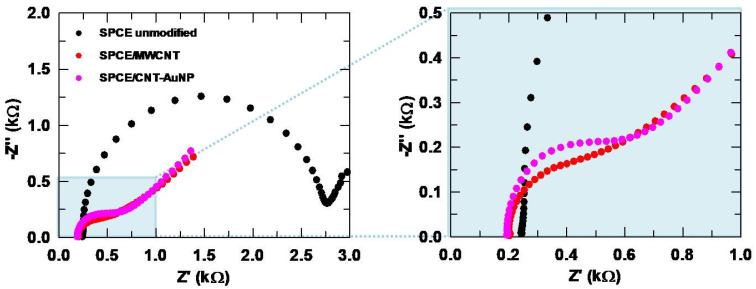
Nyquist plots obtained for the unmodified, and MWCNT and CNT-AuNP modified SPCEs.

**Figure 4 sensors-24-06745-f004:**
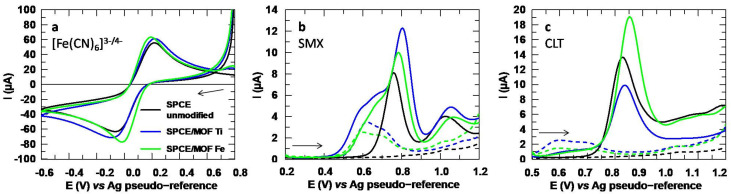
(**a**) Cyclic voltammograms obtained with the MOFs for [Fe(CN)_6_]^3−^ 5 mM. (**b**) DP voltammograms for SMX 151.2 µM and (**c**) CLT 150.0 µM. Electrolytic medium: Na_2_SO_4_ 10 mM at pH values of 6 for [Fe(CN)_6_]^3−^ and 3 for SMX and CLT. Dotted lines in (**b**,**c**) correspond to the blank (Na_2_SO_4_ 10 mM, pH 3).

**Figure 5 sensors-24-06745-f005:**
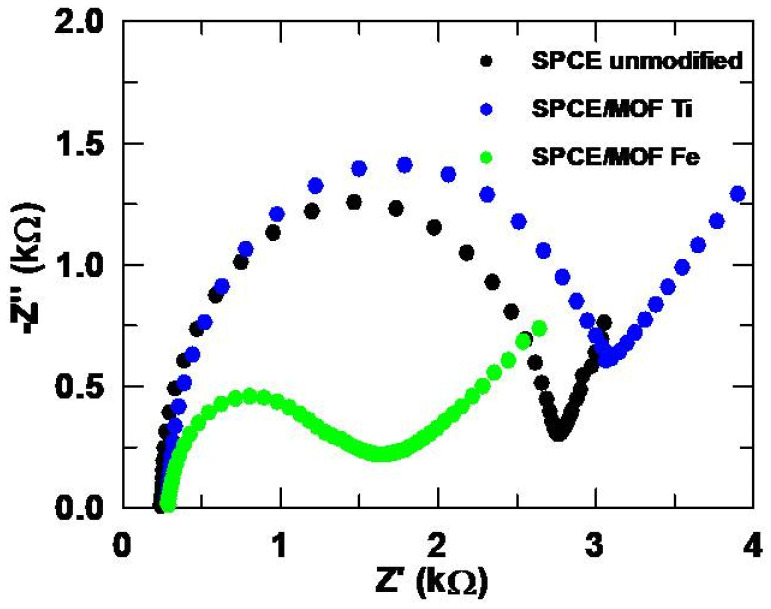
Nyquist plots obtained for the unmodified, and MOF Ti and MOF Fe modified SPCEs.

**Figure 6 sensors-24-06745-f006:**
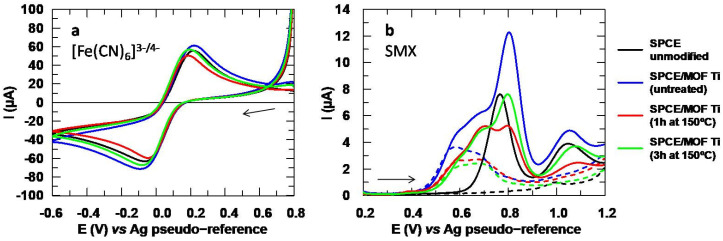
(**a**) Voltammograms obtained with the Ti-MOF after the thermal treatments for [Fe(CN)_6_]^3−^ 5 mM and (**b**) SMX 151.2 µM. Electrolytic medium: Na_2_SO_4_ 10 mM at a pH value of 6 for [Fe(CN)_6_]^3−^ and 3 for SMX. Dotted lines in (**b**) correspond to the blank (Na_2_SO_4_ 10 mM, pH 3).

**Figure 7 sensors-24-06745-f007:**
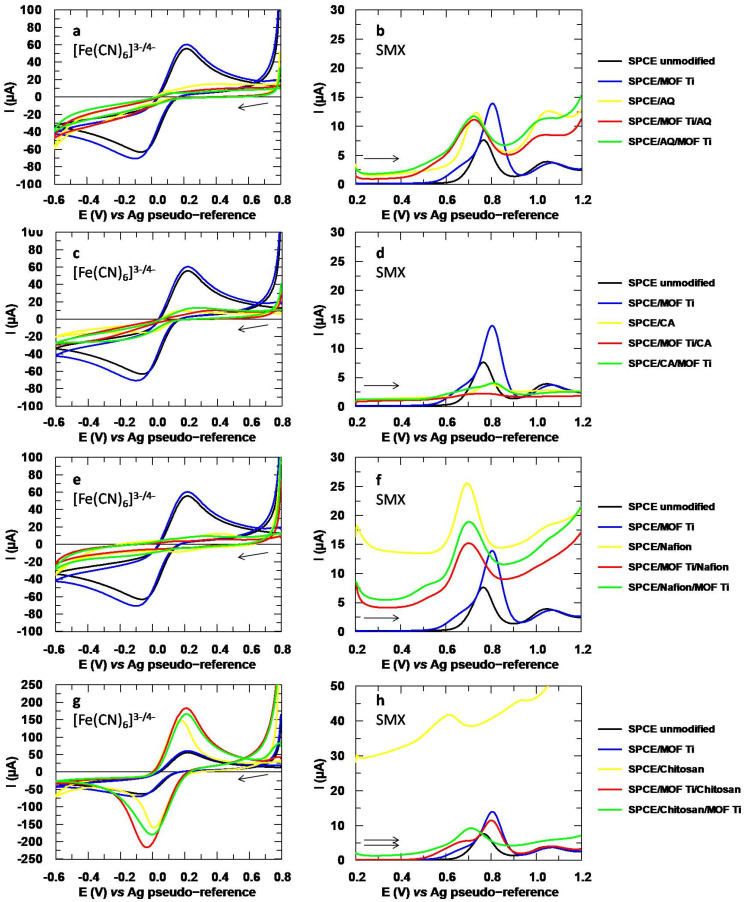
(**a**) [Fe(CN)_6_]^3−^ 5 mM and (**b**) SMX 151.2 µM responses for AQ polymer. (**c**) CA [Fe(CN)_6_]^3−^ and (**d**) SMX responses for CA. (**e**) [Fe(CN)_6_]^3−^ and (**f**) SMX responses for nafion. (**g**) [Fe(CN)_6_]^3−^ and (**h**) SMX responses for chitosan. Electrolytic medium: Na_2_SO_4_ 10 mM at pH values of 6 for [Fe(CN)_6_]^3−^ and 3 for SMX.

**Figure 8 sensors-24-06745-f008:**
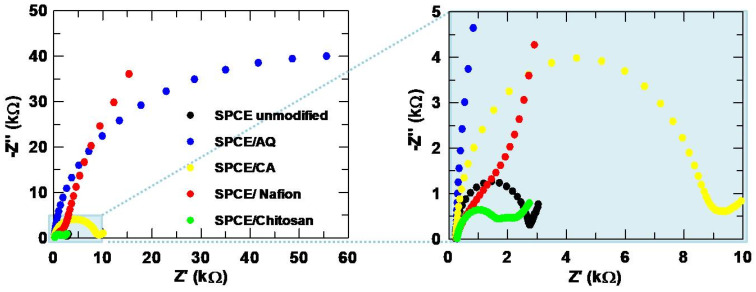
Nyquist plots obtained for the bare and polymer-modified SPCEs.

**Figure 9 sensors-24-06745-f009:**
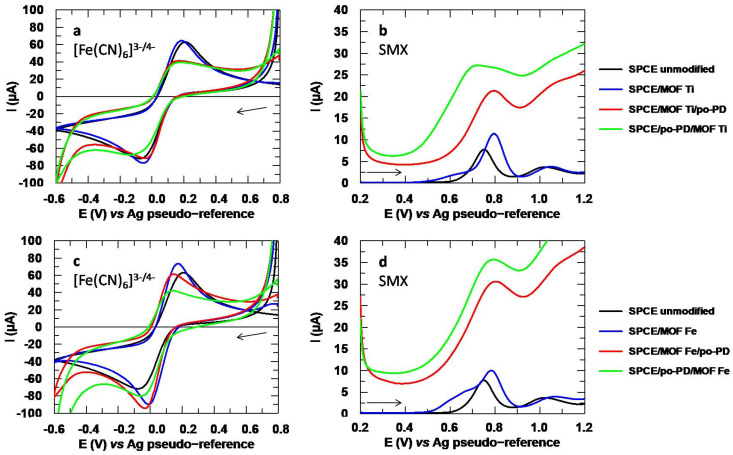
(**a**) Cyclic voltammograms obtained with Ti-based MOF for [Fe(CN)_6_]^3−^ 5 mM and (**b**) DP voltammograms of SMX 151.2 µM. (**c**) Fe-based MOF for [Fe(CN)_6_]^3−^ and (**d**) SMX. Electrolytic medium: Na_2_SO_4_ 10 mM at pH values of 6 for [Fe(CN)_6_]^3−^ and 3 for SMX.

**Figure 10 sensors-24-06745-f010:**
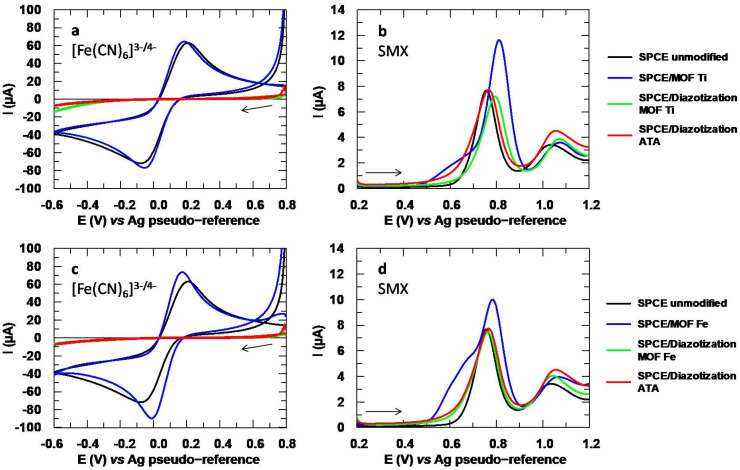
(**a**) Cyclic voltammograms obtained with Ti-based MOF for [Fe(CN)_6_]^3−^ 5 mM and (**b**) DP voltammograms for SMX 151.2 µM. (**c**) Fe-based MOF for [Fe(CN)_6_]^3−^ and (**d**) SMX. Electrolytic medium: Na_2_SO_4_ 10 mM at pH values of 6 for [Fe(CN)_6_]^3−^ and 3 for SMX.

**Figure 11 sensors-24-06745-f011:**
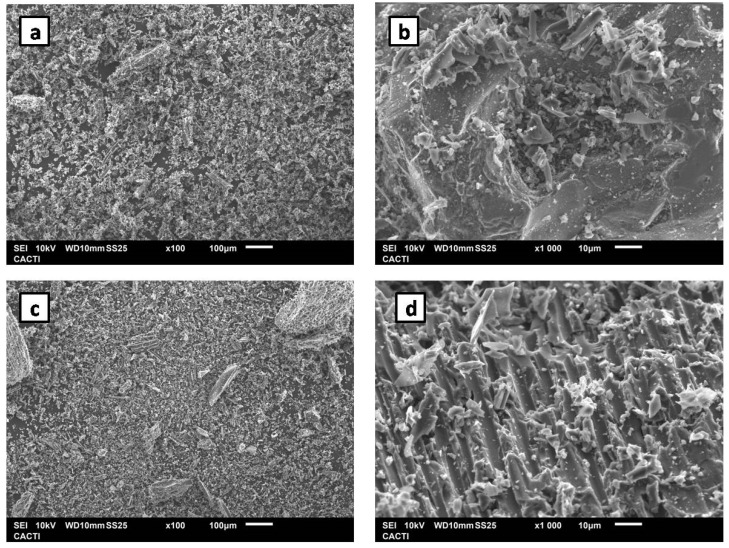
(**a**) SEM images of untreated biochar with a magnification ×100 and (**b**) ×1000. (**c**) Activated biochar with a magnification ×100 and (**d**) ×1000.

**Figure 12 sensors-24-06745-f012:**
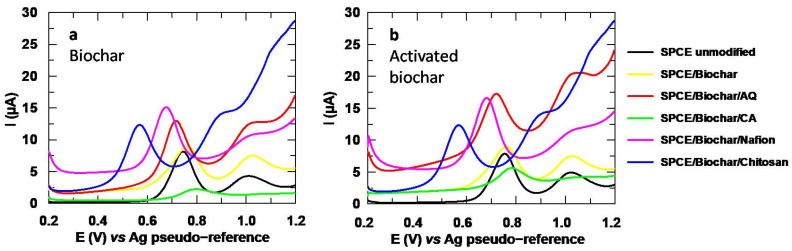
DP Voltammograms obtained for SMX 151.2 µM in Na_2_SO_4_ 10 mM, pH 3 with (**a**) untreated biochar and (**b**) activated biochar.

**Figure 13 sensors-24-06745-f013:**
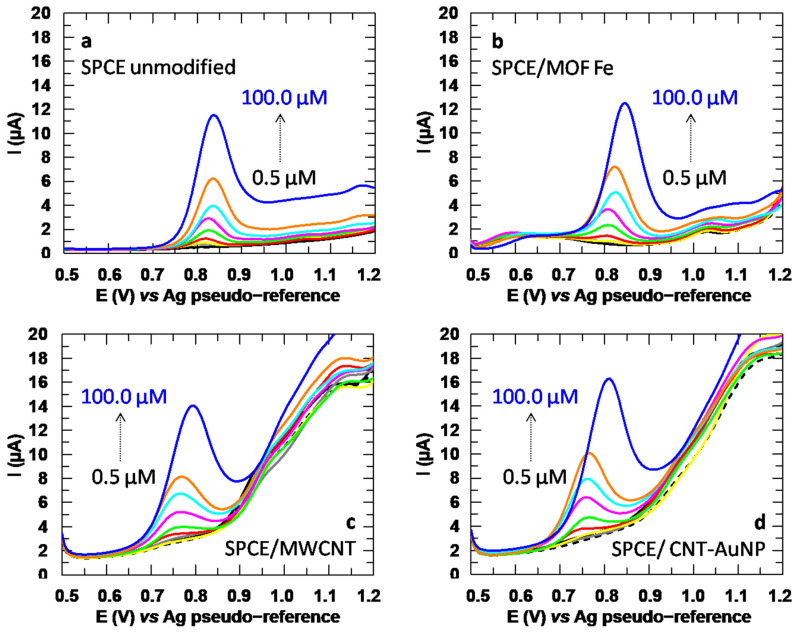
DP Voltammograms obtained for CLT in a concentration range from 0.50 to 100.00 µM with (**a**) unmodified SPCEs, (**b**) SPCEs modified with Fe-MOF, (**c**) SPCEs modified with MWCNT and (**d**) SPCEs modified with CNT-AuNP.

**Table 1 sensors-24-06745-t001:** Data extracted from the main peaks of CLT and SMX with SPCEs modified with the NH_2_-MIL-type MOFs.

Analyte	SPCE	i_p_ (µA)	E_p_ (V)	W_1/2_ (V)	i_p_ Improvement (%)
SMX	Unmodified	6.861	0.759	0.095	-
	MOF Ti	12.442	0.804	0.125	81
	MOF Fe	8.194	0.678	0.268	19
CLT	Unmodified	11.294	0.827	0.093	-
	MOF Ti	8.038	0.832	0.096	−29
	MOF Fe	16.053	0.852	0.087	42

**Table 2 sensors-24-06745-t002:** Data extracted from the voltammograms for SMX with SPCEs modified with the Ti-based MOF and the corresponding polymers.

SPCE	i_p_ (µA)	E_p_ (V)	W_1/2_ (V)	i_p_ Improvement (%)
Unmodified	6.740	0.764	0.100	-
MOF Ti	12.702	0.794	0.104	88
AQ	7.883	0.729	0.102	17
MOF/AQ	7.374	0.719	0.158	9
AQ/MOF	6.434	0.719	0.138	−4
CA	1.732	0.789	0.116	−74
MOF/CA	0.783	0.724	0.216	−88
CA/MOF	2.215	0.814	0.204	−67
Nafion	11.552	0.694	0.107	71
MOF/Nafion	8.047	0.698	0.137	19
Nafion/MOF	9.514	0.698	0.134	41
Chitosan	5.150	0.618	0.159	−24
MOF/Chitosan	9.961	0.799	0.117	48
Chitosan/MOF	5.990	0.709	0.143	−11

**Table 3 sensors-24-06745-t003:** Data extracted from the voltammograms for SMX via electropolymerization and diazotization.

SPCE	i_p_ (µA)	E_p_ (V)	W_1/2_ (V)	i_p_ Improvement (%)
Unmodified	6.801	0.749	0.096	-
MOF Ti	11.962	0.794	0.100	76
MOF Ti/po-PD	8.315	0.779	0.152	22
po-PD/MOF Ti	10.614	0.694	0.246	56
MOF Fe	8.369	0.784	0.160	23
MOF Fe/po-PD	10.112	0.779	0.179	49
po-PD/MOF Fe	10.081	0.759	0.184	48
Diazotization ATA	6.404	0.764	0.117	−6
Diazotization MOF Ti	6.209	0.794	0.108	−9
Diazotization MOF Fe	6.384	0.764	0.112	−6

**Table 4 sensors-24-06745-t004:** Data extracted from the voltammograms for SMX with biochar.

SPCE	i_p_ (µA)	E_p_ (V)	W_1/2_ (V)	i_p_ Improvement (%)
Unmodified	7.036	0.744	0.098	-
Biochar	5.326	0.799	0.123	−24
Biochar/AQ	7.584	0.703	0.110	8
Biochar/CA	3.258	0.769	0.103	−54
Biochar/Nafion	9.901	0.673	0.106	41
Biochar/Chitosan	7.331	0.648	0.116	4
Activated biochar	5.642	0.759	0.119	−20
Activated biochar/AQ	7.617	0.719	0.111	8
Activated biochar/CA	2.574	0.779	0.112	−63
Activated biochar/Nafion	10.152	0.678	0.105	44
Activated biochar/Chitosan	5.259	0.663	0.141	−25

**Table 5 sensors-24-06745-t005:** Calibration parameters for the fitted lines obtained during the studies with the modified SPCEs for CLT.

SPCE	Linear Range (µM)	Intercept (µA)	Slope (µA µM^−1^)	R^2^	LOD (µM)	LOQ (µM)
Unmodified	4.40–100.00	0.22 ± 0.04	0.091 ± 0.001	0.9996	1.32	4.40
MOF Fe	4.81–100.00	−0.03 ± 0.05	0.104 ± 0.001	0.9994	1.44	4.81
MWCNT	6.45–100.00	0.06 ± 0.06	0.093 ± 0.001	0.9993	1.94	6.45
CNT-AuNP	6.36–100.00	0.13 ± 0.07	0.110 ± 0.002	0.9990	1.91	6.36

**Table 6 sensors-24-06745-t006:** CLT content in Etumine^®^ with different SPCEs (declared value: 40 mg CLT/pill).

SPCE	CLT Found (mg)	Recovery (%)	Relative Error (%)	RSD (%)
Unmodified	41.8 ± 0.4	104	4.5	1.0
MOF Fe	43.5 ± 0.4	109	8.8	0.9
MWCNT	42.9 ± 0.4	107	7.2	0.9
CNT-AuNP	43.0 ± 0.8	108	7.5	1.9

## Data Availability

The data presented in this manuscript are available on request from the corresponding author.

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
