# Peer review of "Decorated Electrode Surfaces with Nanostructures and Metal-Organic Frameworks as Transducers for Sensing"

_sensors, 2024, doi:10.3390/s24206745_

Round 1

Reviewer 1 Report

Comments and Suggestions for Authors

The paper investigates the performance of various modified SPCEs for the detection of two model drugs, clotiapine and sulfamethoxazole. Although it has some merit, the manuscript is too long with unnecessary text, poorly organized, and makes overreaching claims. Furthermore, after all the modification investigations, the final results demonstrate that the unmodified SPCE has the lowest LoD and LoQ, which makes all the previous results seem redundant. I cannot recommend the paper for publication in its current form, as it requires extensive re-writing, re-organization, and further meaningful experiments.  

Major Comments

1)    The abstract should supply more detailed information regarding the experimental results. There is much background information but only limited discussion of the actual results.

2)    In Paragraph 1 of the Introduction, you should explain what an SPCE is and give more background information. If readers are not well-versed in SPCEs, they may not understand what the paper is focused on.

3) Similarly, in Paragraph 1 of the Introduction, you should provide more information about how modifiers can improve sensing performance and why that’s important. Generally, the Introduction is limited and does not set the scene well for the manuscript.

4)    In the Introduction, you mention that two model drugs were chosen. Can you provide some context regarding why detecting these drugs is important, particularly outside of a laboratory setting, for which SPCEs are generally intended.

5)   The Reagents and Solutions section is overly wordy and should be more concise to improve its clarity and flow. For example, the text: “For the SPCE modification, carbon nanofibers (98% carbon basis) and magnetic iron oxide nanoparticles (0.5-0.7% iron basis) were used. These modifiers were supplied by Sigma-Aldrich” can easily be merged into one concise sentence.

6)   The text should be more concise throughout the manuscript; too much text is used unnecessarily, which reduces clarity. For example, in Line 167, you state: “Biochar was one of the materials tested as a modifier of the SPCE.” We already know this, and it does not need to be included in the experimental section. There are numerous examples of this unnecessary text throughout the manuscript.

7)    In Lines 317-319, you state: “Therefore, taking into account that three promising modifiers were found for CLT but none that met the all the requirements for SMX, the research focused on this analyte from this point on using the Ti-MOF as modifier.” Firstly, this sentence is ambiguous (e.g., which one does “this” refer to). Secondly, I find it unusual that you would set the study up for two model drugs, abandon one mid-way through the manuscript after achieving some positive results, and then go back to it later. I think you should focus on both throughout the manuscript, as this was the whole point of the paper to compare different modifiers and nanomaterials.

This is the main issue with the manuscript: it is poorly organized, and the justifications for the experiments are unclear. For example, why do you only do the experiments with the concentration range of the drug at the very end? You should have done this from the start, which would have provided meaningful data on how your modifiers impacted sensing performance. Particularly as you conduct numerous experiments to show that nothing improves the SMX detection, and the blank SPCE has the lowest LoD and LoQ for the CLT detection. Therefore, the considerable number of experiments beforehand seems somewhat redundant, especially as you can only investigate sensitivity, linear range, etc. when using a concentration range.

The paper should be simplified and made more impactful. This can be achieved by making the first part of the manuscript significantly more concise and then performing more experiments using concentration ranges of the real drug (and non-targets) to produce meaningful data regarding sensor performance vs. different modifications. You do this for Figure 10, and these results are valuable; this should be the focus of the manuscript. You should do this for both CLT and SMX.

8)  You discuss your modifiers in terms of electrochemical signal; however, you should also discuss them in terms of price and ease of preparation. For example, if one modifier was cheap and easy to prepare but had a 1% worse performance than a more expensive/complex modifier, this would still be the most suitable one for most practical applications. You mention this for biochar but nothing else.

9)   The narrative of the manuscript is overly complicated. You have two model drugs (one used for only some experiments) and many modifiers, polymers, and immobilization methods. It becomes challenging to follow exactly what you are doing in each experiment and why. You should find a way to improve the structure of the manuscript to increase clarity. Tables, flow charts, schematics, or new figures may be useful in achieving this. This also links with Comment 8 regarding the overall manuscript organization and narrative.

10) You state that certain modifications impact your sensor’s “selectivity” (e.g., Line 400). While you see a broadening of peaks in your voltagramms, you have not directly tested selectivity using non-targets; therefore, you cannot state that your selectivity has increased/decreased. The same is true for “sensitivity”; you cannot state that sensitivity is improved just because you have an increased signal for a single concentration. This also links to Comment 8; you should do experiments with a range of concentrations and non-target analytes to allow you to accurately discuss changes to selectivity and sensitivity.

11) There are very few references to other similar studies within the literature. There are numerous similar studies within the literature against which this work should be compared. This would significantly strengthen the Discussion.

12) In Line 527, you state: “Comparing the slopes, the increase in sensitivity achieved with the modifications is clear.” However, as you mentioned, the LoD and LoQ are lower for the blank SPCE. Therefore, why would someone go to the extra effort and cost to produce a modified SPCE if it has a worse LoD/LoQ for the target of interest? I believe this is a key drawback in the work that is overlooked in the Discussions.

13) The discussion of confirming whether the biochar is activated is too long, particularly since most of the associated data/figures are in SI. This should be made much more concise to improve the manuscript quality.

Minor Comments

1)    In Line 40, you state: “analyzing organic compounds thanks to their affinity since all organic molecules have a carbon backbone.” More detail is required to explain this statement.

2)    Informal language is used throughout and should be avoided. For example, “on the other hand” and “stand out” are used repeatedly.

3)  The number of decimal places should be consistent for the same units (e.g., concentration on Lines 119 and 121). This also occurs in other parts of the manuscript.

4)    Cyclic voltammetry and different pulse voltammetry should not be uppercase.

5)    In Line 154, state who the research group is led by: “as well as the research group…”

6)    Figure 1b is discussed in the text before 1a, these should be swapped. Also, the figure caption in Figure 1 appears to be incorrect (1a = SMX not CLT).

7)    The voltagramm figures generally have poor image quality; this should be improved.

8)    In Line 295, you have switched from molar concentration to concentration by weight. Keep consistent units throughout.

9)    You have not mentioned any physiologically relevant concentrations of the two drugs; therefore, no context is provided regarding the concentrations that you are using.

10) In Line 386, the following sentence is mostly redundant: “In light of the presented results, the decision was made to proceed a little further and for this purpose an in situ electropolymerization was carried out.” 

11) The sentence: “There is no need to check the data showed in Table S5 to appreciate that the peaks associated with the redox probe are completely inhibited after diazotization with both the ATA and the MOFs.” is informal and unnecessary.

Comments on the Quality of English Language

1)    The quality of writing and English requires significant improvement throughout the manuscript; please thoroughly proofread and edit the text.

Reviewer 2 Report

Comments and Suggestions for Authors

The manuscript communicated by Pérez and colleagues presents a comprehensive comparison of different functionalizations of nanomaterials and materials on screen-printed carbon electrodes (SPCEs). While the work is valuable and well-executed, there are several major points that need to be addressed before final acceptance:

  1. Scan Rate Studies: The authors claim to have performed scan rate studies, but no data has been provided. Please include the graph showing all scans from 10 to 500 mV/s in the supplementary information (SI).

  2. Title Revision: It would be more precise to modify the title to reflect the use of decorated nanomaterials/materials on the electrodes, rather than using the phrase "Decorated electrode surfaces as transducers for sensing."

  3. SEM Images: SEM images should be provided for all different functionalized materials on the SPCEs. Additionally, please include Electrochemical Impedance Spectroscopy (EIS) data to demonstrate the effects of nanomaterial/material functionalization on the SPCEs.

Minor Comments:

  1. The concentration of [Fe(CN)6]³⁻ should be included in the Materials and Methods section, as it is currently missing.

  2. Clarify the meaning of W1/2 (V) in the manuscript.

  3. Some Differential Pulse Voltammetry (DPV) peaks do not start from zero. Please provide a clear explanation of this phenomenon in the manuscript.

  4. In the voltammograms, the capacitive current appears to be quite high, suggesting that the polymer may not be the most suitable. Specify the extent of this issue using quantitative terms, such as the magnitude of the increase in folds or similar measures.

Comments on the Quality of English Language

Minor english adjustments are required.

Round 2

Reviewer 1 Report

Comments and Suggestions for Authors

-          P1 – Intro: When discussing SPCE, you should highlight their low-cost and ease of production compared to traditional electrode (e.g., GCEs). This is the main benefit of SPCEs, which is the primary focus of the manuscript.

-          Line 54 – Remove ‘since’

-          Line 56 – delete ‘a’

-          Line 69 – Don’t start a sentence with ‘because’

-          Throughout the manuscript, you still make definitive statements about selectivity/sensitivity, which cannot be claimed. You say in your response, that the narrow peak infers better selectivity. As inferring and demonstrating are very different things scientifically, this needs to be clear within the manuscript; particularly as it’s very unlikely that a bare SPCE will be selective for a particular molecule. For example, in the abstract, you state the ‘selectivity was compromised’, which strongly implies you have performed selectivity studies – this is misleading. This occurs throughout the manuscript and the same issues exist with senstivity. You stated several times in the Response document that this is a proof-of-concept study, and further experiments are beyond its scope; therefore, you need to provide complete clarification on what exactly has and has not been performed and what this means about the sensor. You can say that the peaks were narrow, and this may suggest improved selectivity, but that is it. The same is true of sensitivity. I cannot recommend a manuscript for publication until this is properly addressed.

Comments on the Quality of English Language

Mostly fine. 

Round 3

Reviewer 1 Report

Comments and Suggestions for Authors

The authors have addressed my comments and I can recommend the manuscript for publication.